# A protocol to gather, characterize and analyze incoming citations of retracted articles

**Ivan Heibi** [1,2]*, **Silvio Peroni** [1,2]

**1** Department of Classical Philology and Italian Studies, Research Centre for Open Scholarly Metadata, University of Bologna, Bologna, Italy, **2** Department of Classical Philology and Italian Studies, Digital Humanities Advanced Research Centre (/DH.arc), University of Bologna, Bologna, Italy

* ivan.heibi2@unibo.it

## Abstract

In this article, we present a methodology which takes as input a collection of retracted articles, gathers the entities citing them, characterizes such entities according to multiple dimensions (disciplines, year of publication, sentiment, etc.), and applies a quantitative and qualitative analysis on the collected values. The methodology is composed of four phases: (1) identifying, retrieving, and extracting basic metadata of the entities which have cited a retracted article, (2) extracting and labeling additional features based on the textual content of the citing entities, (3) building a descriptive statistical summary based on the collected data, and finally (4) running a topic modeling analysis. The goal of the methodology is to generate data and visualizations that help understanding possible behaviors related to retraction cases. We present the methodology in a structured step-by-step form following its four phases, discuss its limits and possible workarounds, and list the planned future improvements.

**Data Availability Statement:** No datasets were generated or analysed during the current study. All relevant data from this study will be made available upon study completion.

**Funding:** The author(s) received no specific funding for this work.

## Introduction

Retracting a scholarly peer-reviewed article means that the venue that published it (e.g. the journal) withdrawn it, though it may be still reachable and accessible. A retraction is issued through a decision made by the venue's editorial board. The reasons of retraction may be either severe (e.g., data fabrication, plagiarism, or author/s misconduct) or factual (e.g., errors in the research, problems with the research reproducibility, or duplicate publishing). Ideally, a retracted article should not be used as reliable source of information, especially if the reasons for retractions were severe. Pragmatically, this should mean that other articles should not cite and make good use of the methodology, results, or conclusions of a retracted article [1].

In case only a small part of an article reports erroneous data and needs a correction, then the article might go through one or more *partial* retractions. However, if the proposed corrections are not enough to defend the claims of the articles and the journal editor is convinced that the publication is seriously flawed and misleading, then the article can be *fully* retracted. It is worth noticing that, in some circumstances, a full retraction can be done without having a partial retraction of an article in advance. Citations to retracted articles continue also after its

**Competing interests:** The authors have declared that no competing interests exist.

full retraction, even if usually, for different reasons. For instance, while entities that have cited an article before its formal retraction might have been using its methodology or its findings, entities which have cited it after the retraction could have talked about it in a negative connotation and without using it to back a research claim.

In 2009 the Committee on Publication Ethics (COPE) published a document for the retraction guidelines [2]. COPE recommends that the retraction notices should provide sufficient information about the reason for retraction and why the findings are considered unreliable and should explicitly distinguish forms of misconduct from honest error. In addition, retraction notices should be freely available: they should be published and linked to the original article that has been retracted. Indeed, scholars might cite a retracted article unconsciously if such recommendations are not respected. For instance, a study done on the retracted articles of MEDLINE from 1966 to 1997, revealed that almost 94% of the citations to retracted works have been made without having any knowledge regarding their retraction [3].

A citation analysis over one or more retracted article can help us identify facts regarding either a specific case (i.e. one single retracted article) or on the retraction phenomenon related to a particular collection of retracted articles (e.g. those related to a particular scholarly discipline). These kinds of analysis have been largely discussed by scientometricians, and mainly concerned (a) large scale analysis and (b) case of study analysis.

Works belonging to category (a) usually try to answer general questions such as how retractions influence the impact on the authors, institutions and the retracted work itself with the application of an analysis of a single field of study or a broader domain, such as a macro area. Notable examples consist of works such as [4] that used the citation data collected from Web of Science to demonstrate that a single retraction could trigger citation losses through an author's prior body of work. Other negative repercussions on authors and co-authors of retracted articles have been demonstrated also by other works such as [5–7]. Another interesting and recent work done by [8] has defined a multi-dimensional observation framework for retracted publications based on four dimensions: scientific impact, technological impact, funding impact and Altmetric impact. The aim of [8] was to describe and represent the impact of the retracted publications on the scientific community more comprehensively. Beside taking into consideration the entire life cycle of the retracted articles, the authors of [9] for example focused on the analysis of the citations made before the retraction.

Works of category (b) consider either a single or a restricted number of retraction cases (usually, popular ones) and apply their analysis on the entities which have cited the retracted articles taken in consideration. Usually, these studies want to build a general approach to apply on a large-scale scenario starting from the findings they obtain on a restricted number of retraction cases. Notable works part of this category focused on post-retraction citations and analyzed their sentiment toward the retracted article [10], classified the citation contexts [11], or applied a network analysis study [12]. The work done by [13] is a notable example that has combined the features we have just mentioned, in order to demonstrate how retracted research can still gain popularity and how the information environment contributes to such problematic phenomena.

In this paper we took cues from both the kinds of study above and formulated a methodological approach toward the application of a citation analysis starting from an arbitrary number of retracted articles. The method does not have any restriction on the domain of the retracted article/s, neither on the reason for their retraction. The method considers only fully retracted articles–that may or may not have received partial retractions and that:

a. have received the full retraction notice at least a year after their publication year;

b. have received some citations either before or after the retraction year;

c. have been cited by other articles published at least one year after the retraction year.

These constraints are crucial for having a minimal amount of data to enable us analyzing the citing behavior before and after a full retraction event.

The aim of our methodology is to output data and visualizations that help us investigating and inferring behaviors related to the retraction case/s analyzed. In the future, we want to use this methodology for investigating the retraction phenomenon in the humanities domain, which has gained less attention in the past.

In particular, in this paper, we present a methodology that takes one or more retracted articles as input and performs a quantitative and qualitative citation analysis, following four main phases: (1) identifying, retrieving, and extracting basic metadata of the entities which have cited retracted articles, (2) extracting and labeling additional data from the textual content of the citing entities (e.g. abstracts), (3) building a descriptive statistical summary, and finally (4) running a topic modeling analysis. The final outcome of our methodology is an annotated dataset containing a collection of the entities that have cited retracted articles accompanied by a set of features that characterize them, along with a number of charts and dynamic visualizations to help us observe interesting insights from the generated results.

The Section "Methodology" is dedicated to the complete definition of the methodology, based on the four phases we have mentioned above. Then, in Section "Discussion and conclusions", we discuss possible limits and workarounds to overcome them. To this end, we use some of the outcomes of our preliminary experience on the analysis of a specific retraction case: "Ileal-lymphoid-nodular hyperplasia, non-specific colitis, and pervasive developmental disorder in children" by Wakefield et al., published in 1998 [14].

## Methodology

Our methodology is based on four phases, each phase consists of one or more steps as it is summarized in Fig 1, which contains a graphical overview of the methodology workflow (i.e., the four phases), and in Table 1.

The first two phases produce a dataset which stores the entities that have cited the retracted articles taken into consideration. Each entity (i.e. each record in the dataset) will have a set of annotated features (columns). The third phase analyzes the information stored in the dataset (produced and annotated in the first two phases) and summarizes the quantitative aspects into a collection of static charts. Finally, the topic modeling analysis done on the fourth phase takes the textual information (which have been extracted from the full text of the citing entities and stored in the dataset), trains two topic models using this information as input, and builds a set of dynamic visualizations to have an overview on the generated topics and investigate the relation of such topics with the features of the citing entities.

Before executing the methodology, we should select the retracted articles we want to analyze. A useful service that can be used for this purpose, which keeps track and collects retractions of scholarly articles, is Retraction Watch (http://retractionwatch.com/). Using the Retraction Watch database (http://retractiondatabase.org/), we can look for the retracted articles we want to use in our analysis. The methodology we describe here takes in consideration only the articles that have officially received one or more retraction notices and have been fully retracted.

The Retraction Watch database keeps track of several information for each retracted article. In the context of our methodology, we need to keep note of (a) the DOI of the original publication (if any), (b) the year of publication, (c) the authors, (d) the subjects ("History", "Philosophy", etc.) and (e) the year of the retraction notice.

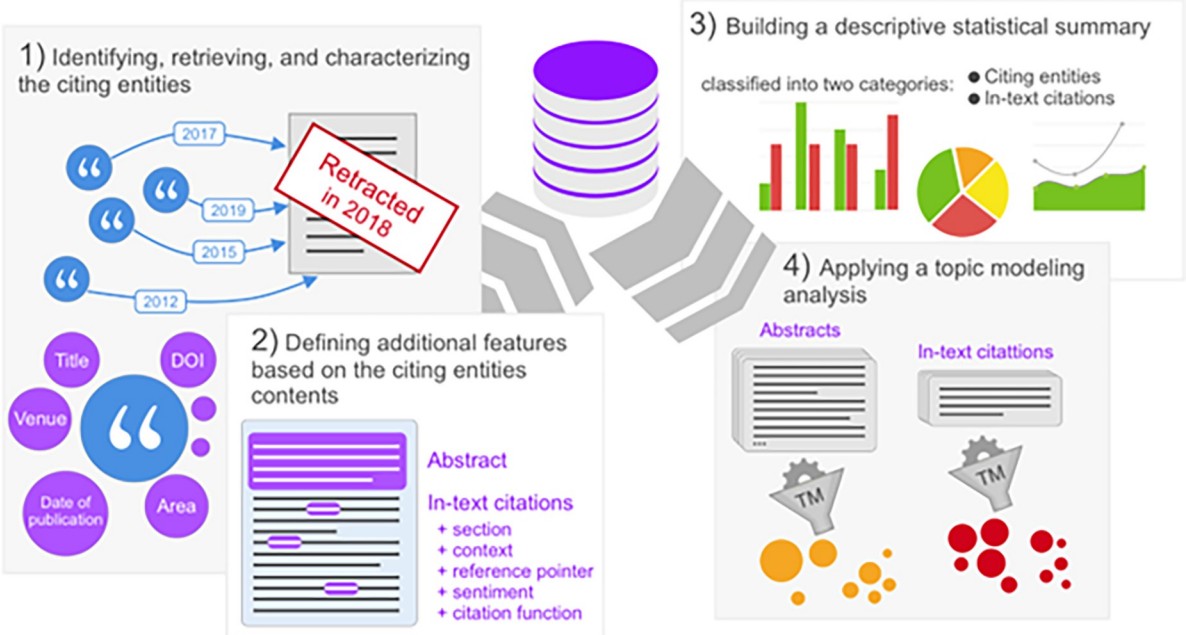

**Fig 1.** A graphical schema representing the methodology in its four phases (form left to right): (1) identifying, retrieving, and characterizing the citing entities, (2) defining additional features based on the citing entities contents, (3) building a descriptive statistical summary, and (4) applying a topic modeling (TM) analysis.

We discuss each phase of the methodology separately in the following subsections. For simplicity, throughout our discussion, we refer to the retracted articles to consider with the abbreviation RET-SET. All the elaborations on RET-SET rely only on open and free services, in order to foster the reproducibility of the methodology. In addition, our approach has no restrictions regarding the adopted language, although this choice should be consistent along the workflow, such that the generated dataset should contain values written in only one specific language (e.g., English).

## Phase 1: Identifying, retrieving, and extracting basic metadata of the entities which have cited a retracted article

Starting from the RET-SET, this phase first identifies all the entities which have cited it and characterize them with their main metadata, such as the title and the year of publication. Then, we check whether any of the citing entities has been or has not been retracted as well, and finally, we classify the citing entities according to their areas of study and specific subject categories, following the Scimago classification (https://www.scimagojr.com/).

The output of this phase is a dataset containing the citing entities (the records) which have cited the RET-SET, with their main metadata (the columns). The same dataset will be further populated with additional data/features on the next phase.

**Step 1.a: Gathering the citing entities.** To get the list of the entities which have cited the RET-SET, we rely on and query only open repositories storing scholarly bibliographic data, to foster the reproducibility of the methodology. Notable examples part of this category are: Microsoft Academic Graph (MAG, https://www.microsoft.com/en-us/research/project/microsoft-academic-graph/) [15], Crossref (https://www.crossref.org/) [16], and OpenCitations (http://opencitations.net/) [17]. OpenCitations provides COCI (https://opencitations.net/index/coci), a citation index which contains details of all the DOI-to-DOI citation links

**Table 1. An overview of the phases of the methodology described in this article.**

| Phase 1: identifying, retrieving, and extracting basic metadata of the entities which have cited RET-SET | Phase 2: extracting additional data and labeling the textual content of the citing entities | Phase 3: building a descriptive statistical summary | Phase 4: running a topic modeling analysis |
|---|---|---|---|
| **Step 1.a: gathering the citing entities:** identifying the list of entities citing the RET-SET and storing their main metadata<br><br>**Input:** RET-SET<br><br>**Output:** For each citing entity: (1.a.1) *DOI*, (1.a.2) *year of publication*, (1.a.3) *title*, (1.a.4) *venue id (ISSN/ISBN)*, (1.a.5) *venue title* | **Step 2.a: extracting textual values from the citing entities:** extracting the citing entities' abstracts and, for each in-text citation (i.e. the location where the citing article cites another work within its content) referencing to the RET-SET, the in-text reference pointer (i.e. the textual device denoting a bibliographic reference such as "[1]"), the textual context of the citation, and the title of the section where it appears | **Step 3.a: analyzing the citing entities:** inferring some descriptive statistics from the data of the citing entities gathered in the previous phases | **Step 4.a: analyzing the abstracts of the citing entities:** automatically extracting the topics through an analysis of the abstracts of the citing entities |
| | **Input:** The citing entities main metadata | **Input:** The dataset produced after phase 2 | **Input:** The abstracts of the citing entities |
| **Step 1.b: marking the retracted entities:** Annotating whether any of the citing entities has been or has not been retracted<br><br>**Input:** The main metadata of the citing entities<br><br>**Output:** For each citing entity: (1.b.1) *is / is not retracted* | **Output:** For each citing entity: (2.a.1) *abstract*, (2.a.2) *in-text citation sections*, (2.a.3) *in-text citation contexts*, (2.a.4) *in-text reference pointers* | **Output:** (3.a.1) a collection of charts to summarize the descriptive statistics of the citing entities | **Output:** (4.a.1) a dataset containing the N most important keywords of each topic, (4.a.2) a dataset containing a list of all the documents of the corpus and their representativeness against each topic, (4.a.3) a collection of dynamic visualizations to observe and investigate the topic modeling results |
| **Step 1.c: classifying the citing entities into subject areas and subject categories:** classifying the citing entities into areas of study and specific subjects, following the Scimago classification (https://www.scimagojr.com/) | **Step 2.b: annotating the in-text citations characteristics:** annotating the intent (why an article is cited, e.g. because it *uses a method* defined in the cited article) and sentiment (positive, neutral, negative) of each in-text citation, and specifying whether the text of the citation context mentions the retraction of the cited article | **Step 3.b: analyzing the in-text citations:** inferring some descriptive statistics from the data of the in-text citations gathered in the previous phases | **Step 4.b: analyzing the in-text citation contexts:** automatically extracting the topics through an analysis of the in-text citation contexts |
| | **Input:** In-text citations and their contexts | **Input:** The dataset produced after phase 2 | **Input:** The in-text citation contexts |
| **Input:** The venues of the citing entities<br><br>**Output:** For each citing entity: (1.c.1) *subject area*, (1.c.2) *subject category* | **Output:** For each in-text citation: (2.b.1) *citation intent*, (2.b.2) *citation sentiment*, (2.b.3) *retraction is / is not mentioned* | **Output:** (3.b.1) a collection of charts to summarize the descriptive statistics of the in-text citations | **Output:** (4.b.1) a dataset containing the N most important keywords of each topic, (4.b.2) a dataset containing a list of all the documents of the corpus and their representativeness against each topic, (4.b.3) a collection of dynamic visualizations to observe and investigate the topic modeling results |

For each phase (column), we list its corresponding steps (cells). Each step is accompanied with a brief description, the inputs needed, and the expected output.

retrieved by processing the open bibliographic references available in Crossref [18]. It also provides a free APIs service to query and retrieve the COCI data at http://opencitations.net/index/coci/api/v1. Crossref makes available a REST API service (https://api.crossref.org) that exposes the metadata that Crossref members (i.e. publishers) deposit in it, such as basic descriptive metadata of publications, funding data, license information, full-text links, ORCIDs, reference lists, etc. Users can search and filter publication information contained in Crossref, that are returned by the API in JSON (a popular open standard file format to store and transmit data objects based on attribute–value pairs). MAG is a knowledge graph which contains the scientific publication records, citations, authors, institutions, journals, conferences, and fields of study. MAG provides a free REST API (https://www.microsoft.com/en-us/research/project/academic-knowledge/) to search, filter and retrieve such data.

These are only some notable examples, additional services (which are open) could be integrated and used in this phase. The choice of the most suitable services relies on the nature of our analysis. For instance, if we are only interested in collecting the citing entities having a DOI value, we might decide to use only COCI which stores only DOI-to-DOI citations. We need to make sure that the adopted service can return the list of entities which have cited a given set of articles (i.e., the RET-SET) and that some basic metadata of such citing entities can be retrieved, in particular: (a) the DOI value, (b) the year of publication, (c) the title of the article, (d) the ID (ISSN/ISBN) of the publication venue, and (e) the title of the publication venue. For instance, the COCI API provides a specific operation, i.e. http://opencitations.net/index/coci/api/v1#/metadata/, which takes a DOI of an entity as input and returns its main metadata, including the citing and cited DOIs of such entity.

If any of the collected entities is either a bibliography, retraction notification, a presentation, or data repository, then it should not be considered in the analysis. Indeed, these items have a limited (or a totally insignificant) impact for the quantitative and qualitative analysis of the methodology described in this article, and it might also produce noise in the data and results, e.g., a retraction notification document always cite the retracted article, although it must not be included in the analysis with the other citing entities. Apart from these publication types that are excluded from the analysis, the methodology takes into consideration several publication types–journal articles, books, book chapters, conference papers, pre-prints, thesis, editorials, etc.

**Step 1.b: Marking the retracted entities.** We look over all the citing entities of the dataset and manually check them in the Retraction Watch database using their DOIs, to mark them as retracted if included in such a database. In case a citing entity does not have a corresponding DOI, we search for any other metadata attribute, e.g., the title or the author. The corresponding entity will be marked with "yes" only if it has received a full retraction notice.

**Step 1.c: Classifying the citing entities according to subject areas and subject categories.** At this step we want to annotate the subject areas and subject categories of each citing entity in the dataset. To do this we consider the titles and IDs (either ISSN or ISBN) of the venues and classify them into specific subject areas and subject categories using the Scimago Journal Classification (https://www.scimagojr.com/). The Scimago Journal Classification groups the journals into 27 main subject areas (medicine, social sciences, computer science, etc.) and 313 subject categories (for medicine, psychiatry and anatomy, for social sciences, law and political science, etc.). These values define two different levels: (1) a macro layer for the subject area, and (2) a lower layer for the specific subject category. We first focus on the journal articles, and then we analyze books/book chapters.

We map each venue of the dataset having an ISSN value into its corresponding area and category following the values specified in the Scimago journal classification. This process is done in a semi-automatic way. If we have an ISSN associated to a citing entity, this check can be automated by matching such value with the venues index of Scimago. Otherwise, we need to manually look for the venue title using the Scimago Journal Rank service at https://www.scimagojr.com/. In case that journals are assigned with more than one subject area or subject category, we will take into consideration all these values. In case we did not find any corresponding value in Scimago for a specific venue, then we search for it using an external source (such as JSTOR at https://www.jstor.org), we annotate it with only one major subject area which is the most consistent with our findings, and we use the same value of the subject area followed by the postfix "(miscellaneous)" (following the Scimago schema) to annotate the subject category.

We also need to classify books and book chapters. We use the ISBNDB service (https://isbndb.com/) to look up for the related Library of Congress Classification (LCC, https://www.loc.gov/catdir/cpso/lcco/). Then, we map the LCC categories found into Scimago subject areas as follows:

1. We consider only the starting alphabetic segment of the LCC code and find the corresponding LCC discipline using a pre-built lookup index [19]. For instance, "RC360" → "RC" → "Medicine".

2. We check whether the value of the LCC subject matches the exact value of a Scimago area using a pre-built Scimago mapping index [19]. If the corresponding value is present, we annotate the subject area with such a value, while the subject category will have the same value with the addition of suffix "(miscellaneous)". In case no corresponding Scimago area is found, we continue to point 3, otherwise we stop.

3. We check whether the value of the LCC subject is a Scimago category using the same pre-built Scimago mapping index. If the corresponding value is present, we annotate the corresponding category with such value, while the area will have the same value used in the Scimago classification to denote the macro area of such a category. In case no corresponding Scimago category is found, we continue to point 4, otherwise we stop.

4. The ISBNs that were not processed in the previous steps need to be manually annotated through the consultation of the complete LCC index (http://www.loc.gov/catdir/cpso/lcco/). In case we did not find any entry in the LCC index for the corresponding ISBN value, we continue to point 5, otherwise we stop.

5. We manually search for the entity using any of its available metadata (e.g., title) and we annotate its subject area with one major category (the one we found to be the most suitable). The subject category is annotated with the same value followed by the word "(miscellaneous)".

## Phase 2: Extracting and labeling additional data from the textual content of the citing entities

This phase requires accessing the full text of the citing entities. In case the full text is not accessible at all, e.g., due to paywalls restrictions, the corresponding entity will still be part of the dataset, but it will not take part in some of the quantitative analysis of the third phase (such as those related to the in-text citations), and it will be completely excluded from the textual analysis of the fourth phase.

The process is divided in two main steps: (a) extracting the textual values from the full-text of the citing entities, and (b) characterizing the in-text citations. More precisely, once we have access to the full-text of a citing entity, we will first extract the article abstract, then the section title and its type, the in-text reference pointer, and context of its in-text citations to the RET-SET. In the second part of this step, we will characterize each in-text citation with three main features: the intent, the sentiment, and whether the context mentions or does not mention the retraction of the cited retracted article.

**Step 2.a: Extracting textual values from the citing entities.** From the full text of the citing entities, we first need to extract the abstracts. Some of the entities might lack an abstract–for example, if we consider the editorials or some book chapters. In this case the abstract slot will be left as empty. Then, we focus on the in-text citations: for each in-text citation to RET-SET, we extract and annotate its in-text reference pointer, context and the title of the section where it appears in.

We define the in-text citation context as the sentence that contains the in-text reference pointer to the bibliographic reference of any article in the RET-SET (i.e., the anchor sentence), plus the preceding and following sentences. There are some exceptions which need to be handled differently, according to where the in-text reference pointer has been specified:

- In a title of a section–the citation context is the entire title;

- In a cell of a table–the citation context is the entire cell;

- In the first sentence of a section–the citation context is the anchor sentence plus the following sentence;

- In the last sentence of a section–the citation context is the anchor sentence plus the previous sentence.

The sections containing the in-text citation are specified according to their type–using the categories "introduction", "method", "abstract", "results", "conclusions", "background", and "discussion" listed in [20]. These categories are used when the intent of the section is clear from its title, otherwise we use other three residual categories, i.e., "first section", "final section" and "middle section" combined with the original title of the section. If the examined full text of the citing entity is not organized into sections (for instance, when we consider some editorials), then the value of its in-text citation section is set to "None". The section title to associate with an in-text citation should always be the heading of the first-level section, even if the in-text citation appears in a lower subsection. For instance, if a citation occurs inside the Section 2.1, then the heading of Section 2 will be considered.

**Step 2.b: Annotating the in-text citations characteristics.** The characteristics we want to analyze are inferred from the observation of the citation context of each in-text citation gathered in the previous step. We examine each in-text citation retrieved and annotate:

- the author's sentiment regarding the cited entity of the RET-SET;

- whether at least one citation context of a particular citing entity does explicitly mention the fact that the cited entity has been retracted, i.e., the citation context contains the word "retract" or one of its derivative words–"retractions", "retracted", etc.;

- the citation intent (or citation function), defined as the authors' reason for citing a specific article (e.g., the citing entity *uses a method* defined in the cited entity).

To specify the citation sentiment, we follow the classification proposed in [10]. Thus, we annotate each in-text citation with one of the following values:

- *positive*, when the retracted article was cited as sharing valid conclusions, and its findings could have been also used in the citing study;

- *negative*, if the citing study cited the retracted article and addressed its findings as inappropriate and/or invalid;

- *neutral*, when the author of the citing article referred to the retracted article without including any judgment or personal opinion regarding its validity.

To annotate the citation intent, we use the citation functions specified in the Citation Typing Ontology (CiTO, http://purl.org/spar/cito) [21], an ontology for the characterization of factual and rhetorical bibliographic citations. Although an in-text citation might refer to more than one citation function at the same time, we annotate each in-text citation with one citation function only. This decision has been taken in order to avoid possible ambiguities when analyzing these values.

To annotate the in-text citation intent we use a decision model which serves as a guiding scheme for the annotator, as it is represented in Fig 2. This decision model is based on a priority ranked strategy that works as follows:

1. we match each in-text citation against at least one of the three macro-categories, i.e. "Reviewing...", "Affecting..." and "Referring..." (first row in Fig 2);

| | Reviewing and eventually giving an opinion on the cited entity<br><br>*Fill in the sentence:*<br>*"My statements are __HEADER__ the cited entity, such that they __CiTO-citation-function__ "*<br><br>*E.g. "My statements are Not on the same page with the cited entity, such that they critiques "* | | | Affecting either the content of or the perception toward the cited/citing entity<br><br>*Fill in the sentence:*<br>*"My statements __CiTO-citation-function__ the cited entity , and affect the content of/perception toward the __HEADER__ "*<br><br>*E.g. "My statements corrects the cited entity , and affect the content of/perception toward the Cited entity "* | | Referring to the cited entity for material/conceptual purposes<br><br>*Fill in the sentence:*<br>*"The document I am citing represents a __HEADER__ , such that my statements __CiTO-citation-function__ the cited entity"*<br><br>*E.g. "The document I am citing represents a General source , such that my statements cites for information the cited entity"* | | |
|---|---|---|---|---|---|---|---|---|
| | **Consistent with** | **Inconsistent with** | **Talking about** | **Cited entity** | **Citing entity** | **Material** | **Concept** | **General source** |
| **10** | (0.1) supports<br>(0.2) confirms | (0.1) derides<br>(0.2) ridicules<br>(0.3) refutes<br>(0.4) critiques | | | | | | |
| **20** | (0.1) agrees with | (0.1) disagrees with<br>(0.2) disputes | | (0.1) compiles<br>(0.2) retracts<br>(0.3) replies to<br>(0.4) speculates on<br>(0.5) corrects<br>(0.6) extends | (0.1) uses data from<br>(0.2) uses method in<br>(0.3) uses conclusions from<br>(0.4) obtains support from | | | |
| **30** | | | (0.1) parodies<br>(0.2) qualifies<br>(0.3) credits | (0.1) updates | (0.1) obtains background from | | | |
| **40** | | | (0.1) discusses<br>(0.2) describes | | (0.1) includes quotation from | | | |
| **50** | | | | | (0.1) includes excerpt from<br>(0.2) documents<br>(0.3) reviews | (0.1) cites as metadata document<br>(0.2) cites as data source<br>(0.3) cites as source document | (0.1) cites as authority<br>(0.2) cites as evidence<br>(0.3) cites as potential solution<br>(0.4) cites as recommended reading<br>(0.5) cites as related | (0.1) cites for information |
| **Score** | 1 | 2 | 3 | 4 | 5 | 6 | 7 | 8 |

**Fig 2. The decision model for the selection of a CiTO citation function to use for the annotation of the citation intent of an examined in-text citation based on its context.** The first large row contains the three macro-categories: (1) "Reviewing . . .", (2) "Affecting . . .", and (3) "Referring . . .". Each macro-category has at least two subcategories, and each subcategory refers to a set of citation functions. The first row defines the suitable citation functions for it with the help of a guiding sentence to be completed according to the chosen sub-category and citation function.

2. for each macro-category we have selected in (1), we choose one or more citation functions from those provided by CiTO;

3. in case we select only one citation function, then we annotate the in-text citation intent with such a value; otherwise

4. we calculate the priority of each citation function we have selected by summing its value in parenthesis (from 0.1 to 0.6) with the corresponding value in the y-axis (from 10 to 50) and in the x-axis (from 1 to 8) shown in Fig 2. The smaller the sum, the more priority the citation function has. For instance, the priority of the citation function "confirms" is 11.2 that is higher than the one of the citation function "describes", which is 43.2. Finally, we select the citation function that has higher priority and annotate the in-text citation function with it.

## Phase 3: Building a descriptive statistical summary

As a result of the first two phases of our methodology, we produce a dataset containing all the entities which have cited the RET-SET accompanied by their gathered information (basic metadata, textual content, citation intent, etc.). In Table 2, we show a summary of all the features contained in the dataset, accompanied by some examples. Based on the collected data, the phase presented in this section generates a set of charts which represent a descriptive overview of all the entities gathered.

Before presenting each step of this phase, we need to clarify the terms we use and that support the overall organization of this phase. In our methodology, we define and refer to four distinct events (happening in a specific year):

- *E-RetPub*, i.e., the publication of the retracted article;

**Table 2. A summary of all the features included in the dataset, generated after the execution of the first two phases of our methodology.**

| Phase 1) Identifying, retrieving, and extracting basic metadata of the entities which have cited RET-SET | | Phase 2) Extracting additional data and labeling the textual content of the citing entities | |
|---|---|---|---|
| **Step 1.a** | | **Step 2.a** | |
| **DOI**: the DOI of the citing article | *None* | **abstract**: the abstract of the citing article | *"In this article, we show the results of a quantitative and qualitative … "* |
| **year of publication**: the year of publication of the citing article | *2021* | **in-text citation section**: the kind of section in the citing entity which includes the in-text citation, taken from the list in [20] | *Introduction* |
| **title**: the title of the citing article | *A qualitative and quantitative citation analysis toward retracted articles: a case of study* | **in-text citation context**: the textual context in the citing entity which includes the in-text citation | *". . . to those introduced in the latter set of studies. In particular, we want to focus on a highly cited retracted article, i.e. [1], that suggested . . ."* |
| **venue id (ISSN/ISBN)**: the ID (ISSN/ISBN) of the venue of publication of the citing article | *1588–2861* | **in-text reference pointer**: the string representing the in-text reference pointer to the retracted article | *[1]* |
| **venue title**: the title of the venue of publication of the citing article | *Scientometrics* | | |
| **Step 1.b** | | **Step 2.b** | |
| **is / is not retracted**: a yes/no value depending on whether the citing article has ("yes") or has not ("no") received at least one retraction notification. *no* | *no* | **citation intent**: the citation intent of each in-text citation in the citing entity, according to the citation functions defined in CiTO | *uses data from* |
| **Step 1.c** | | **citation sentiment**: the sentiment of each in-text citation in the citing entity, classified as positive/negative/neutral, conveyed by the citation context of an in-text citation | *neutral* |
| **subject area**: the subject areas of the venue of publication of the citing article, based on the Scimago Journal Classification (https://www.scimagojr.com/) | *+ Computer Science + Social Science* | **retraction is / is not mentioned**: a yes/no value that indicates if at least one of the citation contexts of the citing article explicitly mentions ("yes") or does not mention ("no") the fact that the cited entity is retracted | *yes* |
| **subject category**: the subject categories of the venue of publication of the citing article, based on the Scimago Journal Classification (https://www.scimagojr.com/) | *+ Computer Science Applications + Library and Information Sciences + Social Sciences (miscellaneous)* | | |

The steps of the two phases are mentioned in separated cells. Each step adds a list of features in the dataset (the first column of the inner tables in each cell). An example of the expected value is given next to each feature (the second column of the inner tables in each cell).

- *E-PR*, i.e., its (possible) first partial retraction;

- *E-FR*, i.e., its full retraction;

- *E-LastCit*, i.e., the last citation it received.

  In addition, we define the following five periods by combining the events mentioned above:

- P0, either [E-RetPub, E-PR] or [E-RetPub, E-FR]–from the year of publication of the retracted article to the year before the one of the first retraction notice, i.e., either from E-RetPub (inclusive) to E-PR (exclusive) or from E-RetPub (inclusive) to E-FR (exclusive) in case the retracted item does not have a partial retraction or if the partial retraction date coincides with the full retraction date;

- P1, [E-PR, E-PR]–the same year of the first partial retraction;

- P2, [E-PR, E-FR]–from the year after the first partial retraction to the year before the full retraction, i.e., between E-PR and E-FR, exclusive;

- P3, [E-FR, E-FR]–the same year of the full retraction;

- P4, [E-FR, E-LastCit]–from the year after the full retraction to the year of the last citation received by the retracted article, i.e. from E-FR (exclusive) to E-LastCit (inclusive).

For the rest of this paper, we use the annotation PERIOD-SET when referring to the above set of periods. Any retracted item in the RET-SET belong to one of these two categories of retracted items: (RET-A) those that have received one partial retraction at least one year before being fully retracted, and (RET-B) those which have received only a full retraction with no partial retractions. The retracted items in RET-B do not have P1 and P2 specified and P0 is set as [E-RetPub, E-FR]–i.e. its PERIOD-SET is equal to {P0, P3, P4}.

Each citing entity have two corresponding values:

- $P_{CIT}$ is the sequences of the inclusive years of the period $P_X$ (among P0-P4) in which the citing entity has been published;

- $P_{CUT}$ is a number between -1 and +1 which identifies if the publication date of the citing entity (and, consequently, the date of the citation it does to the retracted article) is closer to the left margin of $P_{CIT}$ (i.e. when $P_{CUT}$ is close to -1) or if it is closer to the right margin of $P_{CIT}$ (i.e. when $P_{CUT}$ is close to +1).

Considering the years $[Y_1, Y_2, \ldots, Y_i, Y_{i+1}, \ldots, Y_f]$ in $P_{CIT}$, the formula to compute $P_{CUT}$ is defined as follows:

$$P_{CUT}(citing) = \frac{(citing_{date} - Y_1) - (Y_f - citing_{date})}{|P_{cit}| - 1}$$

In practice, the $P_{CUT}$ of a *citing* entity is equal to the difference of the distance in years between the publication date of the citing entity and the first year in $P_{CIT}$ and the distance in years between the publication date of the citing entity and the final year in $P_{CIT}$, over the number of years in $P_{CIT}$ minus one. It is worth noticing that if the publication year of the citing entity is less than the publication year of the cited retracted article, then $citing_{date}$ should be rounded up to the publication year of the cited retracted article. While rare, this situation may happen when the citing article cites the retracted one and the latter has not been formally published yet–but indeed its text was available as, for instance, a preprint.

For instance, let us consider a citing entity A published in 2010 that cites a retracted article R published in 2002, which has received a partial retraction notice in 2008 and a full retraction notice in 2012. Since the publication date of A is included in P2, $P_{CIT}$ (A) is [2009, 2010, 2011] and $P_{CUT}$(A) is 0 (i.e., 0 /2), meaning that the citation that A does to R happened in the middle of P2. Similarly, if another citing entity B published in 2009 cites R, then $P_{CIT}$ (B) is equal to $P_{CIT}$ (A) and $P_{CUT}$ (B) is -1 (i.e., -2/2), meaning that the citation that B does to R happened in the initial part of P2. It is worth mentioning that the computation of $P_{CUT}$ is possible only for a $P_{CIT}$ that includes more than one year. Thus, according to the formula $P_{CUT}$ is undefined in P1 and P3 and when the $P_{CIT}$ calculated in P0, P2, and P4 contains only one year. In these cases, we assign the value 0 to $P_{CUT}$, practically, it means that the corresponding citations are collocated in the middle of the periods.

Using this approach, we can calculate $P_{CIT}$ and $P_{CUT}$ for each citing entity we have collected. These two metrics let us treat all the citing entitles consistently independently from

their specific publication dates and the other dates related to the retracted articles they cite. Thus, we outline a general picture of the entire PERIOD-SET by mapping all the citing entities using the $P_{CIT}$ and $P_{CUT}$ values.

In this phase we analyze separately (as two independent entities) the citing entities and their in-text citations.

**Step 3.a: Citing entities.**    We want to analyze the distribution of the citing entities as a function of two features: (D1) the period $P_X$ characterizing $P_{CIT}$ and the related $P_{CUT}$, and (D2) their subject areas. These distributions are built as a function of the values in the PERIOD-SET. The results should also highlight, visually, the citing entities that have/have not mentioned the retraction.

Considering D1 and the periods in PERIOD-SET, we count the number of citing entities for each fifth in the [-1.0, +1.0] interval characterizing the $P_{CUT}$ values (i.e., [-1.0, -0.61], [-0.60, -0.21], [-0.20, 0.20], [0.21, 0.60], [0.61, 1.0]), and classify them into those that have/have not mentioned the retraction. For instance, a citing entity having $P_{CUT}$ equal to 6/7 and $P_X$ equal to P4 is classified in D1 in the [0.61, 1.0] slice (since 6/7 is equal to 0.857) of the period P4. It is worth mentioning that the choice of splitting the [-1.0, 1.0] interval in balanced fifth is subjective. In principle, according to the analyzed data, one can even decide to split interval in less or more groups.

The visualization created using these values should have a similar format to the one in Fig 3. The results are sketched using a grouped bar chart which represents the number of entities as a function of their $P_{CUT}$ and $P_X$ values. The visualization has two versions according to the category of the retracted articles, i.e. RET-A and RET-B mentioned above. Those having a partial retraction (i.e., RET-A) have all the periods P0-P4 plotted in the chart (i.e., the PERIOD-SET is equal to [P0, P4]), while those that received only a full retraction notice with no partial retractions (i.e., RET-B) have only the periods P0, P3, and P4 plotted in the chart (i.e., the PERIOD-SET is equal to {P0, P3, P4}). In case the RET-SET contains retracted articles from both types, then each type is treated separately and has its own visualization.

The visualization also highlights the citing entities that have/have not mentioned the retraction using a green bar and a red bar, respectively. Of course, the indication of this last feature could be applied only if we have successfully accessed the full text of such entities. To highlight the entities for which we do not have a corresponding full text for, we use a gray bar. The line chart on top of the bar chart of the periods shows the absolute number of citing entities in each slice. The values of the periods P1 and P3 are not part of the line chart, since these values do not follow the [-1.0, 1.0] intervals defined for the other periods.

To represent the distribution of the citing entities among the subject areas we use a pie chart graphic, as in Fig 4. We show the 10 most representative subject areas (each one represented by a color). The rest of the subject areas are grouped under one category "Other subject areas". In addition, we show the percentage of entities that have/have not mentioned the retraction (using the same colors of Fig 3), along with the absolute number of entities part of each different subject area. A similar visualization must be generated for each period in the PERIOD-SET. In case RET-SET contains retracted articles of the two types (i.e., RET-A and RET-B), then this visualization is repeated for each type.

**Step 3.b: In-text citations.**    Considering the in-text citations and their related data, we want to analyze the distribution as a function of three features: (D1) the period $P_X$ characterizing $P_{CIT}$ and the related $P_{CUT}$ of the citing entities containing the in-text citations, (D2) citation intent of each single citation, and (D3) the section where the in-text citation is contained. The distributions are divided according to the periods in the PERIOD-SET, as we have shown in the previous subsection. These distributions are accompanied with the value of the citation sentiment.

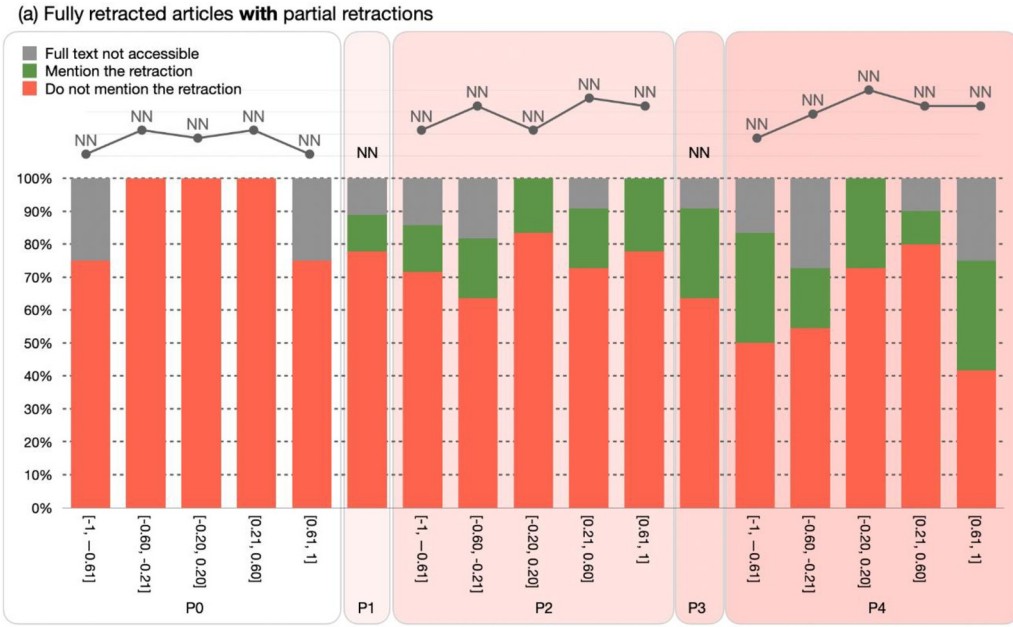

**Fig 3. A graphical representation for the distribution of the citing entities in the PERIOD-SET.** The graphic has two different versions sketched according to the retracted articles categories, i.e. RET-A (A) and RET-B (B). It also highlights the citing entities that have/have not mentioned the retraction, along with the citing entities that do not have an accessible full text.

Fig 5 shows how to represent D1. The visualization is based on a grouped bar chart and the number of in-text citations as a function of the $P_X$ and $P_{CUT}$ of the related citing entities for which we can access their full text. As for the citing entities, the visualization has two versions based on the retracted articles category. In case the RET-SET contains retracted articles of type RET-A and RET-B, then each type is treated separately and has its version. The in-text citations are classified under the three citation sentiments: *negative*, *neutral*, and *positive*, indicated

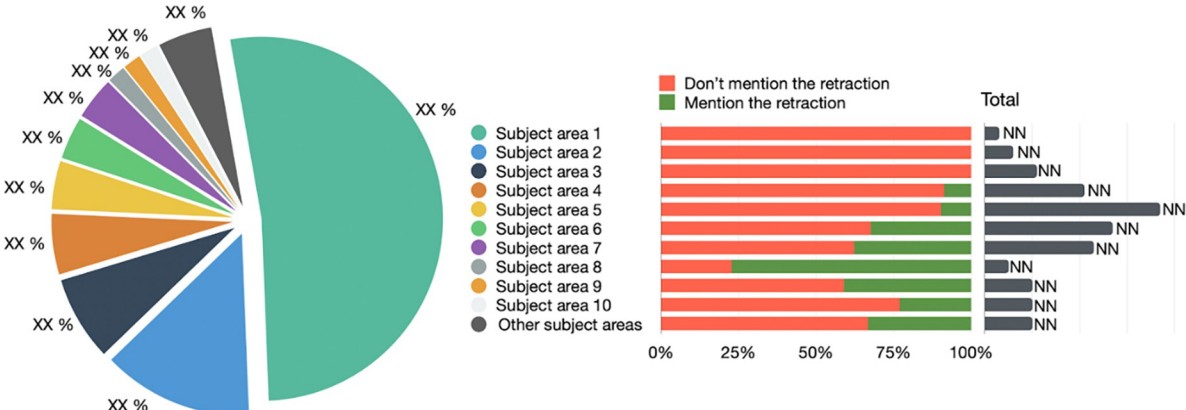

**Fig 4. A pie chart used to represent the distribution of the citing entities across the subject areas.** The chart shows the 10 most representative subject areas and groups the rest under the "Other subject areas" category. The graphic shows also the absolute number of entities for each category, along with the percentages of entities which have/have not mentioned the retraction.

in red, yellow and green, respectively. In addition, we also display the absolute number of in-text citations for each period in the same way we do for citing entities, as described in the previous subsection. In particular, the periods P0, P2, and P4 are characterized by means of fifth slices in the [-1.0, +1.0] interval, while the periods P1 and P3 are represented using one single bar chart and their values are not part of the line chart.

We use a horizontal bar chart to plot D2 and D3, as shown in Figs 6 and 7, respectively. The bars in the chart are grouped considering the sentiment (i.e., negative/neutral/positive) and use the same colors adopted in D1. The length/percentage of the bars is proportional in relation to the total number of in-text citations of related to a particular slice. The absolute numbers are listed next to the chart. Both the visualizations in Figs 6 and 7 need to be built for each period of the PERIOD-SET and refers only to the citing articles for which we can access their full text. For instance, in Fig 6, *citation function 4* represents almost 14% of the total in-text citations of that period, and it contains around 3% of the negative in-text citations of that same period. Fig 6 plots only the section values listed in [20], the other values are part of a separated group labeled "unclassified". In case RET-SET contains retracted articles of RET-A and RET-B, then both the visualizations are repeated for each type.

## Phase 4: Running a topic modeling analysis

A topic modeling analysis is a statistical modeling for automatically discovering the *topics* (represented as a set of words) that occur in a collection of documents. The approach we propose is not restricted to a specific language. A standard workflow for building a topic model is based on three main steps: tokenization, vectorization, and topic model (TM) creation.

Tokenization is the process of converting the text into a list of words, by removing punctuations, unnecessary characters, and stop words. This operation let the topic model analysis focus on the main concepts, by omitting the words that do not have a meaningful impact on the interpretation of the generated topics. In this step, we can also decide to lemmatize and stem the extracted tokens. The lemmatization converts words from third person to first person and verbs in past and future tenses to present. On the other hand, stemming reduces the words to their root form. The aim of these two operations is to reduce inflectional forms and the derivative forms of a word to a common base form.

Then, we create vectors for each of the tokens retrieved. On this step, we can choose between two main models: (a) the term frequency-inverse document frequency (TF-IDF)

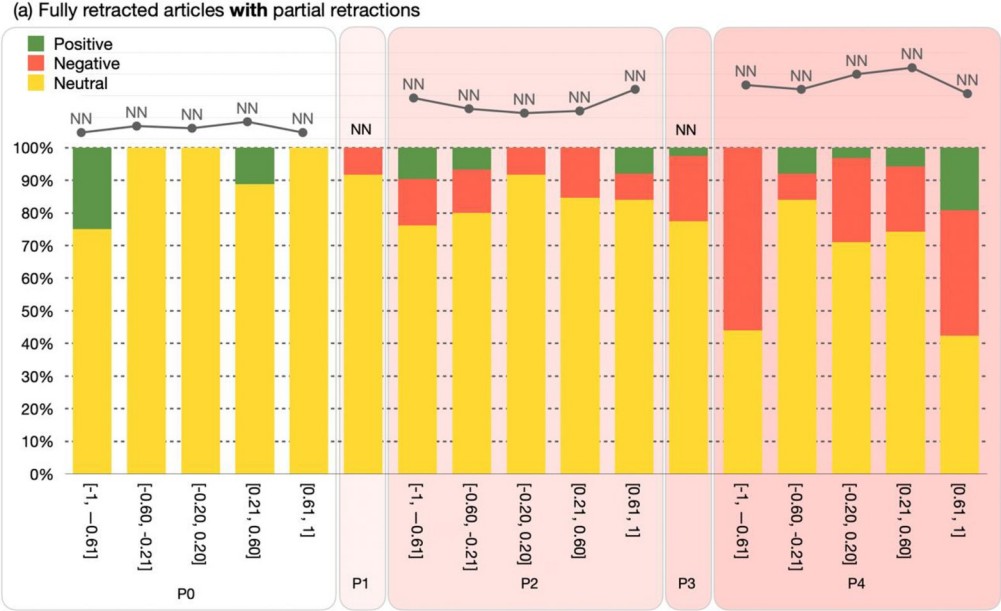

**Fig 5. A graphical representation for the distribution of the in-text citations in the PERIOD-SET.** The periods P0, P2, and P4 are split in fifths, while P1 and P3 are represented using in one slice. The graphic has two different based on the categories of the retracted articles, i.e. RET-A (in the top) and RET-B (in the bottom). The graphic also highlights the neutral, negative and positive in-text citations.

model [22], or (b) a Bag of Words (BoW) model [23]. This choice depends on the type and size of our corpus. Works such as [24] and [25] have investigated this aspect. On the one hand, the authors in [24] suggested that BoW is an appropriate choice when we have to represent the most frequent words but not always the most informative. On the other hand, TF-IDF is a better choice for general purpose cases and when the moderately frequent terms may not be representative of document topics. One of the main findings in [25] is the fact that LDA accuracy depend on the size of the vocabulary and TF-IDF works better with a larger one.

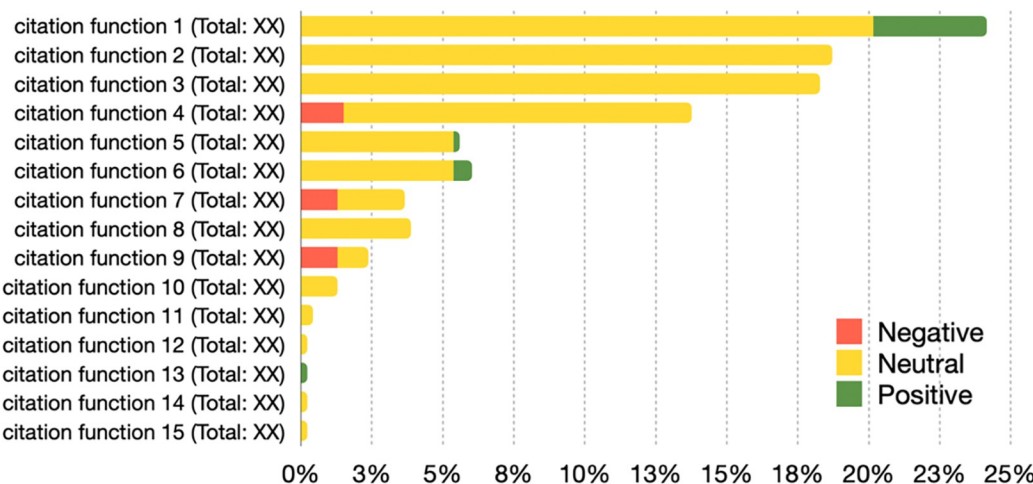

**Fig 6. A horizontal bar chart representing the distribution of the in-text citations according to their citation functions.** The graphic highlights the negative/neutral/positive percentages of in-text citations and mentions, between brackets, the total number on in-text citations annotated with such a citation function. The length (total percentage value) of the bars is in relation to the total number of in-text citation for the period in the PERIOD-SET shown by the graph.

Finally, we build the topic model using the Latent Dirichlet Allocation (LDA) model [26]. As a preliminary step, we need to determine in advance the number of topics to retrieve. Several approaches have been proposed to determine the correct number of topics [27, 28]. A popular approach is based on the value of the topic coherence score, as suggested in [29]. The coherence score is used to measure the degree of the semantic similarity between high scoring words in the topic, and it helps us compare topics that are semantically interpretable from the topics that are artifacts of a mere statistical inference. Throughout this methodology, we decided to adopt this approach. Thus, we calculate the average coherence score for a range of topic models trained with a different number of topics. Then, we plot these values and observe the number of topics for which the average score plateaued, and we select a number of topics indicated in the plateau. For instance, Fig 8 shows the coherence score values of different LDA

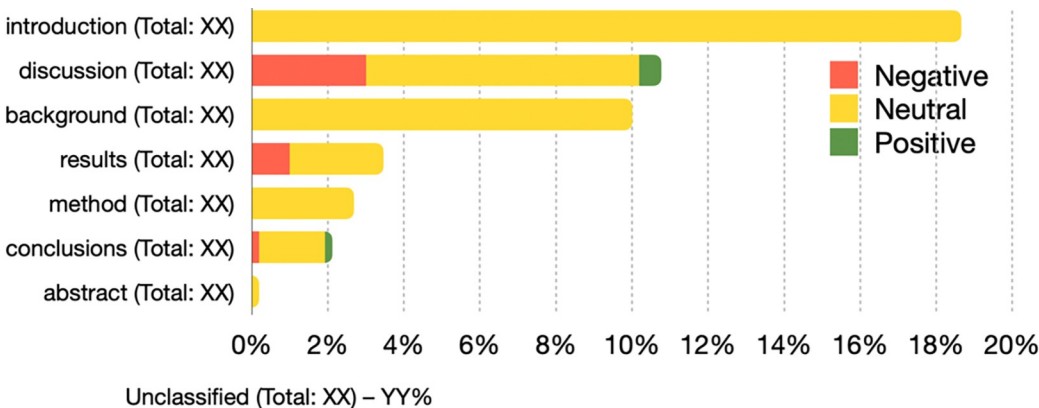

**Fig 7. A horizontal bar chart representing the distribution of the in-text citations according to the section where they appear in.** The graphic highlights the percentages of negative/neutral/positive in-text citations and mentions, between brackets, the total number on in-text citations annotated with such a section. The length (total percentage value) of the bars is in relation to the total number of in-text citation for the period in the PERIOD-SET shown by the graph and is used to sort the values of the sections.

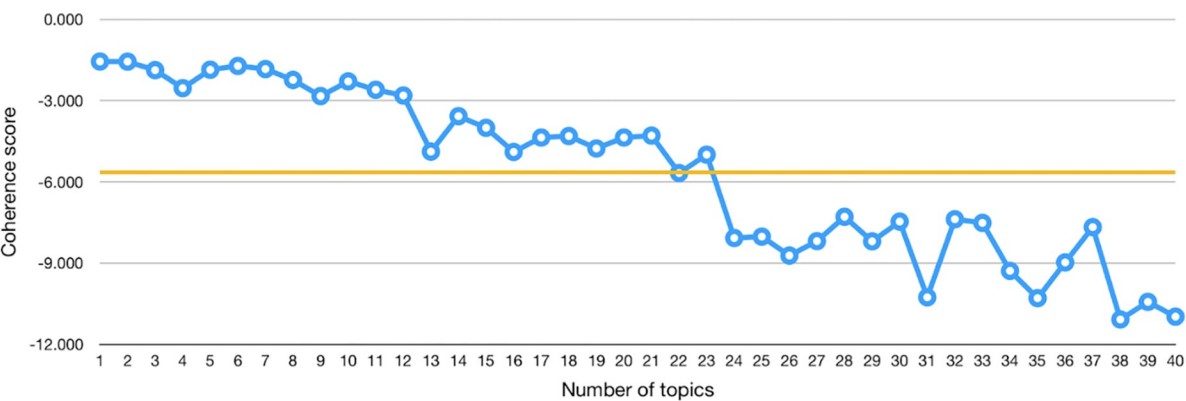

**Fig 8. A plot example of the coherence score of different LDA topic models built using a variable number of topics, from 1 to 40.** The orange line is the average value, and it plateaus around 22–23 topics.

topic models built with a number of topics ranging from 1 to 40. The coherence score plateaued around 22–23 topics. Thus, we might decide to train the LDA topic model with 22 topics.

We build and run the LDA topic model using MITAO [30] (https://github.com/catarsi/mitao), a visual interface to create a customizable visual workflow for text analysis. Using MITAO we generate two datasets:

- the *N* most important keywords of each topic, which represent the *N* most useful and probable terms for interpreting a topic, ranked according to their probability value.

- documents representativeness, i.e., the lists of all the documents of the corpus and their representativeness against each topic.

We use MITAO to also generate two dynamic visualizations which help us gathering new insights from the results of the topic modeling. These visualizations are named LDAvis (Latent Dirichlet Allocation Visualization) and MTMvis (Metadata-based Topic Modeling Visualization).

LDAvis, shown in Fig 9, provides a graphical overview of the topic modeling results [31]. This visualization plots the topics as circles in a two-dimensional plane whose centers are determined by computing the distance between the topics and uses a multidimensional scaling to project the inter-topic distances onto two dimensions. The topic prevalence is represented by the dimension of the area of each circle. In addition, LDAvis shows a global list of 30 terms ranked using the *term saliency* measure. This saliency measure combines the overall probability of a term with its distinctiveness: how informative is a specific term for determining the generation of a topic versus any other randomly selected term [32]. In addition, by selecting a particular topic, LDAvis shows a list of 30 terms ranked using the *relevancy* measure which is used to rank terms within each topic to aid users in topic interpretation activities. This measure is controlled by a weight parameter $\lambda$, which allows one either to rank the terms in a decreasing order of their topic-specific probability if close to 1 or to rank terms solely by their lift if close to 0.

MTMvis, shown in Fig 10, is a dynamic and interactive visualization that shows the representativeness of the topics in the documents based on customizable metadata accompanying these documents.

Fig 11 shows how the topic model workflow is specified in MITAO. The workflow takes a collection of documents (i.e., the node "documents"), builds the LDA topic model (i.e., the "A"

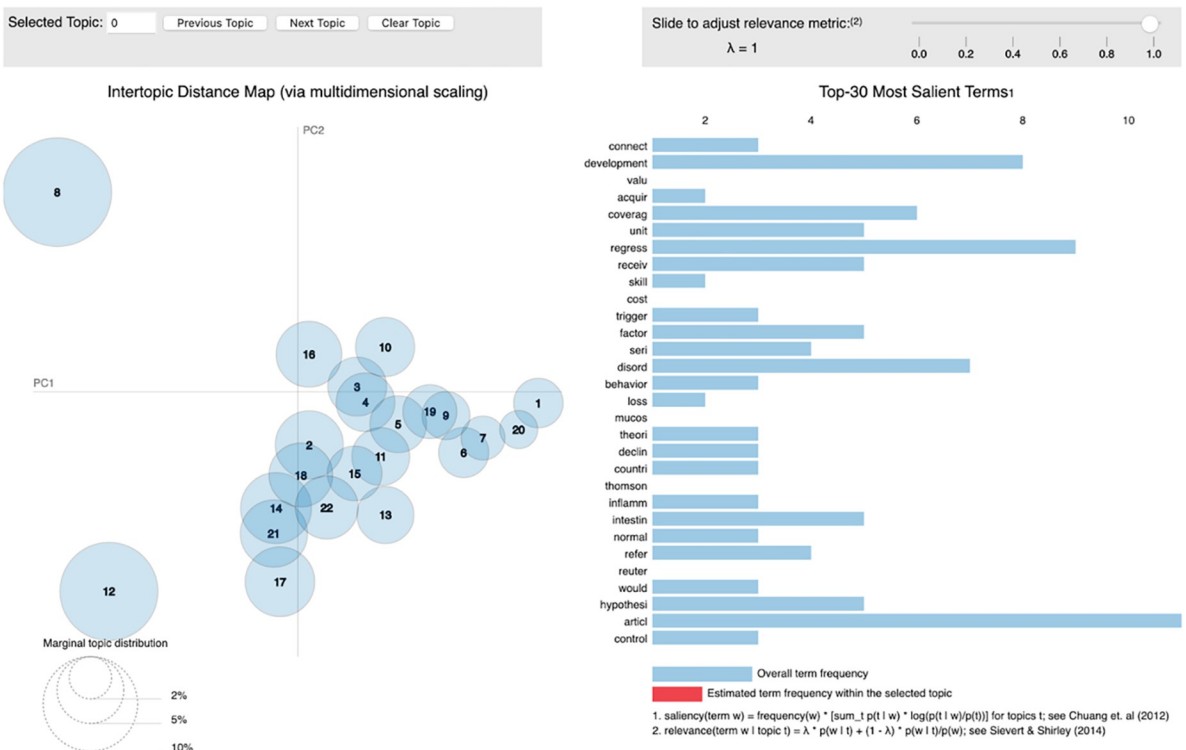

**Fig 9. The LDAvis interface.** The left side of the visualization plots the topics in a two-dimensional plane whose centers are determined by computing the distance between topics. On the right side LDAvis lists 30 terms ranked using the *term saliency* measure, this list might show the 30 terms ranked using the *relevancy* measure of a specific topic if selected from the left graphic.

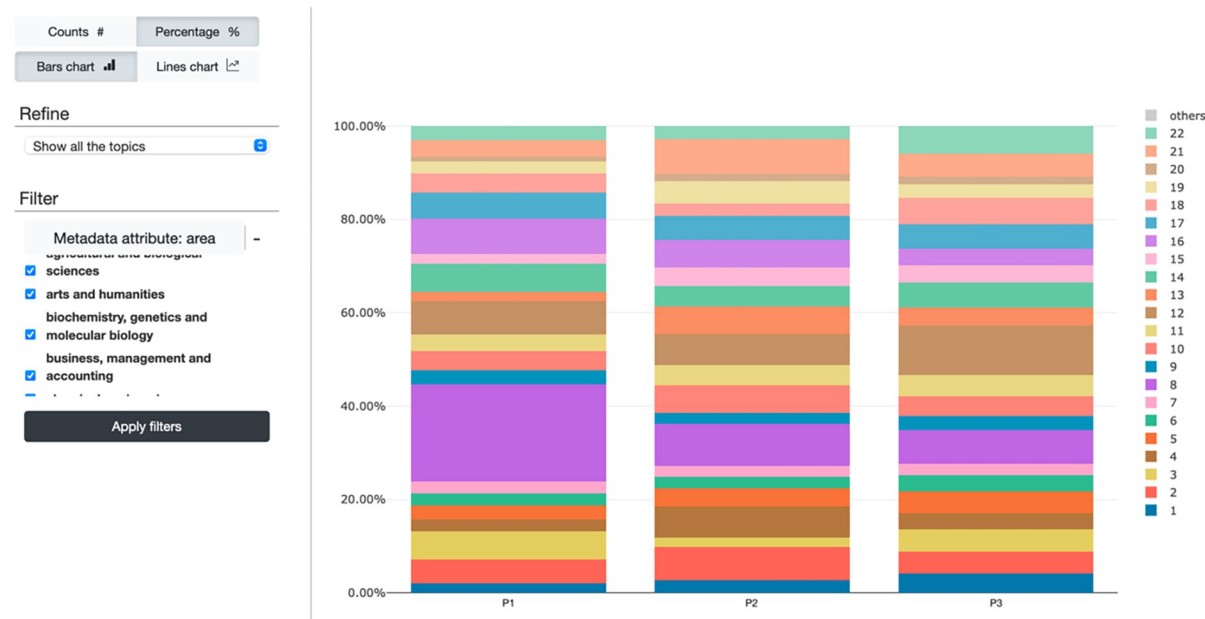

**Fig 10. The MTMvis interface.** On the left side, users can modify some visual and filtering parameters to dynamically change the main visualization. Each topic is colored differently. The chart plots the topics as a function of an established metadata attribute (X-axis values), e.g., the PERIOD-SET.

rectangle in red), generates the two datasets (i.e., the "B" rectangle in blue), and builds the two visualizations LDAvis and MTMvis (i.e., the "C" rectangle in green) using the metadata of the original documents (i.e., the node "meta").

We apply a topic modeling analysis on the abstracts of the citing entities and on their in-text citation contexts citing articles in RET-SET. Each analysis follows the process we have discussed above, with some adaptations depending on the initial document corpus to use (article abstracts and in-text citation contexts).

**Step 4.a: Abstracts.**   In this case, the "documents" in Fig 11 represent the collection of the abstracts of all the citing articles that cites RET-SET. Citing entities with no abstract, due to the scenarios mentioned in Section "Step 2.a: extracting textual values from the citing entities" (e.g., editorials), are not part of the analysis. Each document contains one abstract, and the total number of documents considered in the creation of this topic model is equal to the number of citing entities for which we were able to retrieve the abstract. In the tokenization process, we need to remove the stop words (such as "the", "they", and "you") and other words which are very common in abstracts and cause noise in the obtained results, such as "background", "summary", "method", "results", etc. Finally, we build two MTMvis graphs on top of the two metadata attributes: (a) PERIOD-SET, and (b) subject area. In case RET-SET contains retracted articles of type RET-A and RET-B, then these visualizations are done for each type.

**Step 4.b: In-text citation contexts.**   In this case, the "documents" in Fig 11 represent the in-text citation contexts where a citation to RET-SET is contained. Each document contains the context of one in-text citation, and the total number of documents considered in the creation of this topic model is equal to the total number of in-text citation contexts we were able to retrieve. Citing entities with no in-text citations, because their full text was inaccessible, are not part of this analysis. In the tokenization process, we need to remove the stop words and other additional words which are part of the bibliographic reference of the cited retracted articles, e.g., name of the author (s) or the title of the publication cited. Finally, we build two MTMvis graphs using the same metadata attributes: (a) PERIOD-SET, and (b) subject area. In case RET-SET contains retracted articles of type RET-A and RET-B, then these visualizations are done for each type.

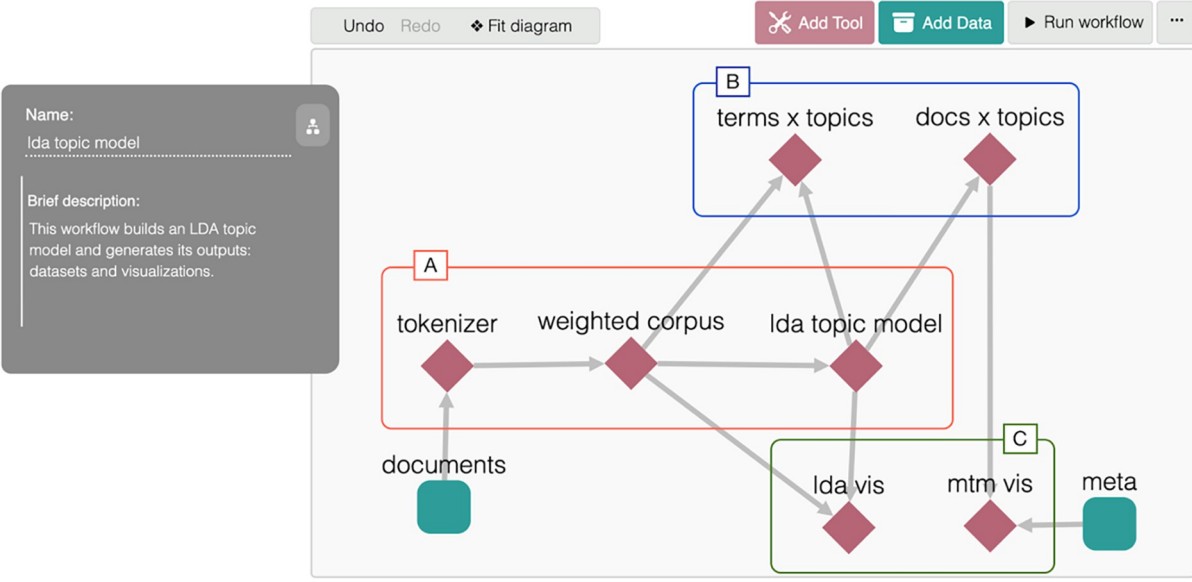

**Fig 11.** The MITAO workflow used for building a LDA topic model (i.e., rectangle A) and generating the datasets (rectangle B), and the visualizations (rectangle C). The workflow takes two inputs: the documents, and the metadata of the documents.

## Discussion

The methodology we have presented in this article has been devised after long studies of methods, results, and experience gained from our previous works, such as [33] which focused on a citation analysis of a popular and highly cited retracted article "Ileal-lymphoid-nodular hyperplasia, non-specific colitis, and pervasive developmental disorder in children" by Wakefield et al. published in 1998. Even if conducted on a single retracted article, such a study gave us an overview of the nature of the data and the common issues that may arise with a citation analysis on retracted papers. In this section, we will discuss some important aspects and limits that our methodology may have, providing some insights and examples taken from [33].

Our methodology has no restrictions on the number of retracted articles to take as input, contrary to the analysis done on [33] which is based only on one retraction case. This flexibility was essential, since we plan on reusing this methodology on a larger scale and especially for the investigation of the retraction phenomenon in the humanities domain, which gained less attention in the past. In addition, the methodology described in this article allows us also to consider and group the analysis according to two distinct retraction scenarios, i.e., when partial retractions either are or are not characterizing the life of a retracted article. A consistent quantitative and qualitative analysis on the entities citing the RET-SET, despite of the heterogeneity in the years when the various events (i.e., publication date, partial retraction date, retraction date, last citation date) involving the RET-SET happened, is possible by calculating the two values $P_{CIT}$ and $P_{CUT}$ for each citing entity. Using these two values we can classify the citing entities under the correct slice according to each period in PERIOD-SET.

We perform two separated quantitative and qualitative analysis (and generate their visualizations) following the retracted items type (i.e. RET-A and RET-B). Although, in theory, RET-A and RET-B could be combined together if we analyze the periods they have in common (i.e. P0, P3, and P4), we think that having two distinct analysis for RET-A and RET-B is the best choice since, currently, we do not know whether having either partial retractions or only the full retraction has the same effect on the citation behaviors in P0.

The various phases of the methodology may be run by combining automated and computational approaches with manual elaboration. Our plan is to clearly devise and describe which approach (i.e. computational vs manual) to use to run the various steps of the methodology in a forthcoming and separated document that will be published in Protocols.io (https://www.protocols.io/). In particular, the Protocols.io document will include the automatable operations as scripts, while the manual operations will be formalized as a set of conditions which can guide, simplify, and limit the ambiguities of the user on its annotation. In addition, all the data gathered, and the visualization produced will be published in Zenodo (https://zenodo.org/).

The first phase of the methodology can be easily implemented by means of a computational approach, by using scripts which can query OpenCitations' APIs or MAG's APIs. Other services could be adopted as well but we will need to further investigate how to integrate them using additional ad-hoc scripts. Thus, the time of execution of this phase should relatively low.

Instead, marking the citing entities which have been retracted is the higher time-consuming step, this is due to the fact that this step relies on manual querying of the Retraction Watch database using its web search interface. An automatic approach is possible if we can access the complete dataset, and this could be done only by signing an agreement with Retraction Watch and on behalf of an institution. This part has not been considered as a direct part of the methodology, due to its bureaucratical nature.

The subject area and category classification step can potentially introduce some biases, in particular when we did not manage to find the venues of the articles citing RET-SET in Scimago. In these cases, indeed, it is needed a manual subjective interpretation from the

annotator. Although, based on our experience from the previous studies, we found a limited number of venues without a corresponding classification in Scimago.

Another crucial attribute to investigate and consider in this phase, e.g. in possible future adoptions of this methodology, concerns the classification of journals in Q1-Q4 quartiles (an indicator for the journal quality). This information is part of the data provided by Scimago. However, we cannot apply this classification to citing entities of type books/book chapters and, as such, books and book chapters cannot take part of such an analysis that consider the quartiles as shown in Scimago.

Although the reason of retraction is part of the information that we can get from Retraction Watch, this property has not been included in the list of attributes to annotate and has not been considered in the methodology–e.g., to distinguish forms of misconduct from honest error. The choice of considering articles retracted for a particular reason instead of another affects the initial input set of retracted articles to analyze using our methodology and, as such, it should be an explicit choice made by adopters of the methodology who may decide to include some articles (e.g. those retracted for misconducts) instead of others. Due to the variability of these choices, we decided not to track explicitly retraction reasons in the methodology, which can be added anyway if necessary for a specific study.

The second phase is the highest time-consuming phase. It is totally based on a manual processing, consequently, it is prone to human errors, e.g., text typing, text translation, the correct definition of the in-text citation context windows, etc. However, these errors could be detected in the second step of the phase while annotating the characteristics of the in-text citations, since the annotator needs to read the extracted text again. An important limit of this phase is related to its dependency on the availability of the full text of the citing entities, which relies on the user having access to paywall contents. In case of using English as main language for the documents, non-English full texts are also accepted as long as the original text is written in a Latin alphabet language. In this case, the dataset produce by the methodology stores a translated version of the original text done by the annotator or by means of automatic tools such as Google Translate. As a result of this flexibility, the translated text may not accurately represent the original one and we might lose some crucial information. Although, since the topic modeling analysis aim is to infer a collection of meaningful keywords, we think this trade-off is reasonable, since we are not interested in a perfect representation of the text.

From our prior experience, we think that the second step of the second phase, i.e., "Annotating the in-text citations characteristics", the most delicate one. As direct consequence of several studies we have conducted in the past on the manual and automatic identification of citation functions, such as [34] and [35], we have defined a guiding schema (summarized in Fig 2) to help the annotator in the definition of the citation intent. Of course, this process could still be biased by human's subjective judgement while reading and analyzing the in-text citation context. We expect that a future adoption of this guiding schema, which we have already experimented in [33], may give us additional clues to, eventually, refine and extend the proposed schema with additional constraints to limit possible ambiguities. For instance, a particular ambiguous situation we faced in [33] arose when trying to annotate the following in-text citation context: "This study revealed that mental health correlated with religiosity . . .". In that sentence, the word "This" may be referred to either the article itself (the citing one) or another study cited and discussed in the previous sentence. In these cases, for example, we can decide to always apply the second option, or to force a larger definition of the context window. Another aid could come from formalizing the annotation of the citation sentiment into a decisional schema, similarly to what we have done for the citation intent. This may help the annotator in the choice and reduce his subjective interpretations. For example, a possible rule to include in a potential decisional schema is: if the citing article uses the word "controversial" in reference to the cited retracted article, then the citation

sentiment should be marked as *negative*. However, in our methodology, we did not propose any annotation schema for mapping the citation sentiment.

The calculation of $P_{CIT}$ and $P_{CUT}$ should be done automatically to each citing entity. In addition, we think that the Protocols.io document should also include the description of an automatic process for building the charts we have presented in the section describing the third phase of our methodology. We want to ensure that this activity, though, is done using open services and software to foster the reproducibility of the methodology. Our idea is to create a customizable workflow as it has been done in MITAO for what it concerns the fourth phase of the methodology. In addition, a potential improvement would be to convert the static charts into a dynamic format (e.g., dynamically outlined following the definition of some filters).

In the last phase of the methodology (i.e., "Running a topic modeling analysis"), most of the execution is handled by MITAO. However, the preparation of the documents to use as input of the process is up to the user. Our plan is to integrate an additional sub-module in MITAO which takes the annotated dataset as input and automatically generates a collection of documents based on an established criterion. Such extension would allow MITAO to automatically generate the abstracts and in-text citation contexts collections from the dataset generated in the previous phases of the methodology, as well as other documents collections based on different attributes, e.g., the subject area.

It is worth mentioning that a first descriptive version of this methodology has been published on Protocols.io [36]. However, it needs to be redefined and formalized following the phases and elaborations introduced in this paper.

Another potential future improvement to the current methodology would be the integration of a citation network analysis step, e.g., through the introduction of an additional phase to be executed after the generation of the citing entities datasets. For instance, we could use either the retracted article(s) or its citing entities as seed, as suggested by the work of Van der Vet and Nijveen [37], who proved the importance of such analysis in similar research contexts. This analysis can enlighten us on the negative outcomes of the propagation of the retracted research results, enabling us to investigate whether the original retraction had affected the decision of retraction of the citing article. However, we will leave this extension as a future work to test on an appropriate dataset which includes citation chains of retracted articles.

## Author Contributions

**Conceptualization:** Silvio Peroni.

**Data curation:** Ivan Heibi.

**Investigation:** Ivan Heibi.

**Methodology:** Ivan Heibi.

**Resources:** Ivan Heibi.

**Software:** Ivan Heibi.

**Supervision:** Silvio Peroni.

**Visualization:** Ivan Heibi.

**Writing – review & editing:** Ivan Heibi, Silvio Peroni.

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
