## [Decision Letter · Decision Letter 0]

10 Dec 2021

PONE-D-21-18322A protocol to gather, characterize and analyze incoming citations of retracted articlesPLOS ONE

Dear Author (S),

Thank you for submitting your manuscript to PLOS ONE. After careful consideration, we feel that it has merit but does not fully meet PLOS ONE’s publication criteria as it currently stands. Therefore, we invite you to submit a revised version of the manuscript that addresses the points raised during the review process.

We look forward to receiving your revised manuscript.

Kind regards,

Amira M. Idrees, Associate Professor

Academic Editor

PLOS ONE

Journal Requirements:

Additional Editor Comments (if provided):

Reviewers' comments:

Reviewer's Responses to Questions

**Comments to the Author**

1. Does the manuscript provide a valid rationale for the proposed study, with clearly identified and justified research questions?

Reviewer #1: Yes

Reviewer #2: Yes

2. Is the protocol technically sound and planned in a manner that will lead to a meaningful outcome and allow testing the stated hypotheses?

Reviewer #1: Yes

Reviewer #2: Yes

3. Is the methodology feasible and described in sufficient detail to allow the work to be replicable?

Reviewer #1: Yes

Reviewer #2: Yes

4. Have the authors described where all data underlying the findings will be made available when the study is complete?

Reviewer #1: Yes

Reviewer #2: Yes

5. Is the manuscript presented in an intelligible fashion and written in standard English?

Reviewer #1: Yes

Reviewer #2: Yes

6. Review Comments to the Author

You may also provide optional suggestions and comments to authors that they might find helpful in planning their study.

Reviewer #1: Overall, the work is good. Some rooms for improvement are as follows:

1. It is rare to combine discussion and conclusion. It is better to separate both sections, so the study contribution is more comprehensive to be read in Discussion section.

2. Please use title case to write title or subtitle, not the lowercase.

3. Please proofread the paper, and make sure the format is consistent throughout the paper.

Reviewer #2: Authors present a protocol to gather and analyze citations of retracted articles. The topic presented is important to the whole research community since it deals with a serious issue. The manuscript is well-written and the methodology they present is - to some extent - comprehensive. All details in each step are provided extensively. Tables and figures are quite clear.

HOWEVER, it does not contain any kind of data (datasets, inputs, results…), rather, it describes a pure theoretical methodology. Definitely, such manuscripts are accepted in the research community and fulfill PLOS ONE’s terms.

Authors mentioned that they would publish the various steps as scripts and set of conditions for the computational and manual parts respectively in “protocols.io”. And they mentioned also that all data gathered as well as results would be published in “Zenodo” (although they did not mention if they were willing to run the protocol on all retracted articles or just those that belong to the humanity domain as they had mentioned earlier in the same page).

I WOULD run that comprehensive and interesting protocol and gather the results and publish them in one complete paper. This suggestion does not affect my judgment on the manuscript regarding all other aspects.

I have the following suggestions/comments authors may take into consideration:

- On page 2, it is mentioned that “COPE recommends that the retraction notices should provide sufficient information about the reason for retraction and why the findings are considered unreliable and should explicitly distinguish forms of misconduct from honest error”, however, later in the manuscript, such property was not included in the study. Is it because journals – when retracting articles – do not follow this recommendation from COPE so one may say that the data is unavailable?

- In phase 1 – on page 6 – it is stated that “Then, we check whether any of the citing entities has been or has not been retracted as well”. It is important then to check if such cases are considered as a “chain reaction” … did the original retraction heavily affect the decision of retraction of the citing article? Such a question is crucial to be investigated, consequently it should be mentioned in the methodology by explaining the procedure needed to investigate it.

- On the same page, it would be better to describe briefly “JSON”.

- In phase 2 – page 8 - it is stated that “In case the full text is not accessible at all, e.g., due to paywalls restrictions, the corresponding entity will still be part of the dataset, but it will not take part in some of the quantitative analysis of the third phase…”. Do authors mean by “not accessible at all” to include articles published in non-open access journals, however, a preprint or a draft is available somewhere. If the answer is “yes”, what the sentence that contains the citation was removed in the final manuscript?

- A missing part in this study is related to study and analyze the following data:

o How the retracted articles are distributed among the four quartiles defined by Scimago? (percentages)…

o What about the journals that cited them? Same quartile, lower, higher? Is the in-text citation categorized as “positive”, “negative” or “neutral”? In other words, are the majority of journals of higher quartile that cite retracted articles in journals of lower quartile mentioned those paper in a “negative” way?

Based on the above, I would accept the manuscript.

7. PLOS authors have the option to publish the peer review history of their article (what does this mean?). If published, this will include your full peer review and any attached files.

Reviewer #1: No

Reviewer #2: No

---

## [Author Response · Author response to Decision Letter 0]

10 Feb 2022

Dear Editors,

First of all, we would like to thank the responsible editor and the reviewers for their comments. We have addressed them in the paper where appropriate. Please find attached all our answers to reviewers' main concerns.

Reviewer #1

Reviewer’s comment: “1. It is rare to combine discussion and conclusion. It is better to separate both sections, so the study contribution is more comprehensive to be read in Discussion section.”

Our answer: We mixed both sections into one section “Discussion”, following the guidelines at https://journals.plos.org/plosone/s/submission-guidelines which states “ These sections may all be separate, or may be combined to create ... a mixed Discussion/Conclusions section (commonly labeled “Discussion”) ...”

Reviewer’s comment: “2. Please use title case to write title or subtitle, not the lowercase.”

Our answer: Done. The new title is: “A Protocol to Gather, Characterize and Analyze Incoming Citations of Retracted Articles”

Reviewer’s comment: “3. Please proofread the paper, and make sure the format is consistent throughout the paper.”

Our answer: Following the guidelines at https://journals.plos.org/plosone/s/submission-guidelines we changed the manuscript text to be double-spaced and included page numbers and line numbers. Also, the table titles and descriptions have been moved above the tables. We have proofread the paper fixing all the grammatical mistakes found.

Reviewer #2

Reviewer’s comment: “HOWEVER, it does not contain any kind of data (datasets, inputs, results…), rather, it describes a pure theoretical methodology. Definitely, such manuscripts are accepted in the research community and fulfill PLOS ONE’s terms.”

Our answer: Indeed this work describes a theoretical methodology to apply to the analysis of retractions. As also stated in the guidelines of PLOS for “Study protocols” articles: “... Relate to a research study that has not yet generated results”. Thus, at the time of writing, we did not have run any specific study by using this methodology, even if it has been defined by taking inspiration from previous works we have done on the topic.

Reviewer’s comment: “Authors mentioned that they would publish the various steps as scripts and set of conditions for the computational and manual parts respectively in “protocols.io”. And they mentioned also that all data gathered as well as results would be published in “Zenodo” (although they did not mention if they were willing to run the protocol on all retracted articles or just those that belong to the humanity domain as they had mentioned earlier in the same page).”

Our answer: The protocol we have defined in this paper represents a general methodology for the application of a citation analysis on retractions with no restrictions on the domain or the reasons of retraction. As we have stated in the Introduction section: “In this paper we took cues from both the kinds of study above and formulated a methodological approach toward the application of a citation analysis starting from an arbitrary number of retracted articles. The method does not have any restriction on the domain of the retracted article/s, neither on the reason for their retraction”. Indeed, our plan is to apply this protocol to the humanities domain since its close to our research domain, yet the protocol is not limited to that case only. In fact, the current version of the methodology, uploaded on Protocols.io [2], could be appied on any set of retractions given as input. Yet, such protocol still needs extra adjustments in order to be fully compliant with the methodology we have presented. We added a small paragraph in the “Discussion” section to calrify this fact. 

Reviewer’s comment: “I WOULD run that comprehensive and interesting protocol and gather the results and publish them in one complete paper. This suggestion does not affect my judgment on the manuscript regarding all other aspects.”

Our answer: Indeed, this is for sure the upcoming plan, however, we wanted to first formalize and present a theoretical protocol in a separate work to gain feedback on the overall quality of the methodology before applying such protocol on a case study (e.g., retractions in the humanities domain), as requested in the PLoS ONE guidelines for this kind of publication (“Study Protocols”, https://journals.plos.org/plosone/s/submission-guidelines#loc-study-protocols).

Reviewer’s comment: “- On page 2, it is mentioned that “COPE recommends that the retraction notices should provide sufficient information about the reason for retraction and why the findings are considered unreliable and should explicitly distinguish forms of misconduct from honest error”, however, later in the manuscript, such property was not included in the study. Is it because journals – when retracting articles – do not follow this recommendation from COPE so one may say that the data is unavailable?”

Our answer: This is a very good point. The reason for retraction is part of the information that we can get from Retraction Watch which is extracted from the retraction notice published by the journals. So to answer your question: no, that's not the reason for not including such property in the study. Instead, we felt that this property would introduce an additional feature that makes the analysis of the results and the gathering process more complex (since we need to consider all the other values as a function of the retraction reason), that may depend also on the particular disciplines we are referring to in the study or the particular set of retracted article to consider (e.g., in principle, we could select only retracted articles compliant with only one motivation). However, our idea is to adapt and customize the current methodology depending on the particular case study, e.g. by adding a new phase to be run at the very beginning (i.e., Phase 0) to define the retraction case study and the retracted publications we are willing to analyze. This phase should investigate several features including the reason for retraction, but it is out of the scope of the present methodology that should be reuseable in a plethora of different scenarios concerning retractions. Indeed, users decide the retraction cases to include/exclude in the rest of the process based on their own goals, and it is not something that is forced by the methodology presented in this article. Our paper has been modified to address and clarify this point.

Reviewer’s comment: “- In phase 1 – on page 6 – it is stated that “Then, we check whether any of the citing entities has been or has not been retracted as well”. It is important then to check if such cases are considered as a “chain reaction” … did the original retraction heavily affect the decision of retraction of the citing article? Such a question is crucial to be investigated, consequently it should be mentioned in the methodology by explaining the procedure needed to investigate it.”

Our answer: Indeed, this is another crucial interesting aspect to investigate. The best way to investigate this aspect is through a network analysis of the citations, as has been done by other works (e.g., [1]). However, this step strongly depends on the particular set of retracted articles we are considering that, as mentioned in a comment before, is actually provided as input of the methodology, and it is not produced as a consequence of the methodology itself. It would be possible that such input can be structured in a way that presents such chain of retractions, indeed, but it is not necessarily the case in all possible scenarios. Thus, this aspect should be considered in the analysis only of the specific users’ goals wants to consider it in his/her study. Thus, in future application of the methodology, it would be possible and necessary to extend part of the analysis proposed by the methodology to address also this chain of retractions. However, since these situations are peculiar only to particular case studies, we decided not to deal with them explicitly in the methodology. We extended the discussion (i.e., the “Discussion” section) of the paper to clarify this aspect.

Reviewer’s comment: “- On the same page, it would be better to describe briefly “JSON”.”

Our answer: Done. 

Reviewer’s comment: - “In phase 2 – page 8 - it is stated that “In case the full text is not accessible at all, e.g., due to paywalls restrictions, the corresponding entity will still be part of the dataset, but it will not take part in some of the quantitative analysis of the third phase…”. Do authors mean by “not accessible at all” to include articles published in non-open access journals, however, a preprint or a draft is available somewhere. If the answer is “yes”, what the sentence that contains the citation was removed in the final manuscript?”

Our answer: No, we can’t be sure that the preprint (or any available draft) is the same paper that has been published later. The example you have mentioned is indeed one of the main issues that could be faced. Therefore, in the present methodology, if the cited paper is unfortunately not accessible from the publisher’s website, we don’t consider any other substitutive form for that specific cited entity. Of course, this does not prevent a user of the methodology to relax this rule by including preprints, if he/she thinks it is worth of using them.

Reviewer’s comment: “ - A missing part in this study is related to study and analyze the following data: (1) How the retracted articles are distributed among the four quartiles defined by Scimago? (percentages)… (2) What about the journals that cited them? Same quartile, lower, higher? Is the in-text citation categorized as “positive”, “negative” or “neutral”? In other words, are the majority of journals of higher quartile that cite retracted articles in journals of lower quartile mentioned those paper in a “negative” way?”

Our answer: Indeed the journals quartiles is a very important information to annotate and include in the dataset. However, we cannot apply this value to citing entities of type books/book chapters (which are part of the citing entities gathered). This fact will leave out all the non-journal papers from the analysis of the results. Possible adaptions to relax the methodology and include this feature should be considered in the future versions of the methodology. We have extended the “Discussion” section to address this fact.

Best regards, 

Ivan Heibi 

Silvio Peroni

References 

[1] van der Vet, P. E., & Nijveen, H. (2016). Propagation of errors in citation networks: A study involving the entire citation network of a widely cited paper published in, and later retracted from, the journal Nature. Research Integrity and Peer Review, 1(1), 3. https://doi.org/10.1186/s41073-016-0008-5

[2] Heibi, I., & Peroni, S. (2020). A methodology for gathering and annotating the raw-data/characteristics of the documents citing a retracted article v2 [Preprint]. https://doi.org/10.17504/protocols.io.bqqumvww

---

## [Decision Letter · Decision Letter 1]

20 Jun 2022

A Protocol to Gather, Characterize and Analyze Incoming Citations of Retracted Articles

PONE-D-21-18322R1

Dear Dr. Heibi,

We’re pleased to inform you that your manuscript has been judged scientifically suitable for publication and will be formally accepted for publication once it meets all outstanding technical requirements.

Kind regards,

Vijayalakshmi Kakulapati, Ph.D

Academic Editor

PLOS ONE

author addressed all review comments 

the manuscript is  acceptable 

Review Comments to the Author

Reviewer #2: I would like to thank the authors for their detailed responses. They have provided convincing answers and proper adjustments. No further comments.

---

## [Editor Report · Acceptance letter]

23 Jun 2022

PONE-D-21-18322R1 

A Protocol to Gather, Characterize and Analyze Incoming Citations of Retracted Articles 

Dear Dr. Heibi:

I'm pleased to inform you that your manuscript has been deemed suitable for publication in PLOS ONE. Congratulations! Your manuscript is now with our production department. 

Kind regards, 

on behalf of

Dr. Vijayalakshmi Kakulapati 

Academic Editor

PLOS ONE